# Training Confidence-calibrated Classifiers for Detecting Out-of-Distribution Samples

**Kimin Lee**[*]   **Honglak Lee**[§,†]   **Kibok Lee**[†]   **Jinwoo Shin**[*]
[*]Korea Advanced Institute of Science and Technology, Daejeon, Korea
[†]University of Michigan, Ann Arbor, MI 48109
[§]Google Brain, Mountain View, CA 94043

## Abstract

The problem of detecting whether a test sample is from in-distribution (i.e., training distribution by a classifier) or out-of-distribution sufficiently different from it arises in many real-world machine learning applications. However, the state-of-art deep neural networks are known to be highly overconfident in their predictions, i.e., do not distinguish in- and out-of-distributions. Recently, to handle this issue, several threshold-based detectors have been proposed given pre-trained neural classifiers. However, the performance of prior works highly depends on how to train the classifiers since they only focus on improving inference procedures. In this paper, we develop a novel training method for classifiers so that such inference algorithms can work better. In particular, we suggest two additional terms added to the original loss (e.g., cross entropy). The first one forces samples from out-of-distribution less confident by the classifier and the second one is for (implicitly) generating most effective training samples for the first one. In essence, our method jointly trains both classification and generative neural networks for out-of-distribution. We demonstrate its effectiveness using deep convolutional neural networks on various popular image datasets.

## 1 Introduction

Deep neural networks (DNNs) have demonstrated state-of-the-art performance on many classification tasks, e.g., speech recognition (Hannun et al., 2014), image classification (Girshick, 2015), video prediction (Villegas et al., 2017) and medical diagnosis (Caruana et al., 2015). Even though DNNs achieve high accuracy, it has been addressed (Lakshminarayanan et al., 2017; Guo et al., 2017) that they are typically overconfident in their predictions. For example, DNNs trained to classify MNIST images often produce high confident probability $91\%$ even for random noise (see the work of (Hendrycks & Gimpel, 2016)). Since evaluating the quality of their predictive uncertainty is hard, deploying them in real-world systems raises serious concerns in AI Safety (Amodei et al., 2016), e.g., one can easily break a secure authentication system that can be unlocked by detecting the gaze and iris of eyes using DNNs (Shrivastava et al., 2017).

The overconfidence issue of DNNs is highly related to the problem of detecting out-of-distribution: detect whether a test sample is from in-distribution (i.e., training distribution by a classifier) or out-of-distribution sufficiently different from it. Formally, it can be formulated as a binary classification problem. Let an input $\mathbf{x} \in \mathcal{X}$ and a label $y \in \mathcal{Y} = \{1, \ldots, K\}$ be random variables that follow a joint data distribution $P_{\text{in}}(\mathbf{x}, y) = P_{\text{in}}(y|\mathbf{x}) P_{\text{in}}(\mathbf{x})$. We assume that a classifier $P_\theta(y|\mathbf{x})$ is trained on a dataset drawn from $P_{\text{in}}(\mathbf{x}, y)$, where $\theta$ denotes the model parameter. We let $P_{\text{out}}(\mathbf{x})$ denote an out-of-distribution which is 'far away' from in-distribution $P_{\text{in}}(\mathbf{x})$. Our problem of interest is determining if input $\mathbf{x}$ is from $P_{\text{in}}$ or $P_{\text{out}}$, possibly utilizing a well calibrated classifier $P_\theta(y|\mathbf{x})$. In other words, we aim to build a detector, $g(\mathbf{x}) : \mathcal{X} \rightarrow \{0, 1\}$, which assigns label 1 if data is from in-distribution, and label 0 otherwise.

There have been recent efforts toward developing efficient detection methods where they mostly have studied simple threshold-based detectors (Hendrycks & Gimpel, 2016; Liang et al., 2017) utilizing a pre-trained classifier. For each input $\mathbf{x}$, it measures some confidence score $q(\mathbf{x})$ based on a pre-trained classifier, and compares the score to some threshold $\delta > 0$. Then, the detector assigns

label 1 if the confidence score $q(\mathbf{x})$ is above $\delta$, and label 0, otherwise. Specifically, (Hendrycks & Gimpel, 2016) defined the confidence score as a maximum value of the predictive distribution, and (Liang et al., 2017) further improved the performance by using temperature scaling (Guo et al., 2017) and adding small controlled perturbations to the input data. Although such inference methods are computationally simple, their performances highly depend on the pre-trained classifier. Namely, they fail to work if the classifier does not separate the maximum value of predictive distribution well enough with respect to $P_{\mathtt{in}}$ and $P_{\mathtt{out}}$. Ideally, a classifier should be trained to separate all class-dependent in-distributions as well as out-of-distribution in the output space. As another line of research, Bayesian probabilistic models (Li & Gal, 2017; Louizos & Welling, 2017) and ensembles of classifiers (Lakshminarayanan et al., 2017) were also investigated. However, training or inferring those models are computationally more expensive. This motivates our approach of developing a new training method for the more plausible simple classifiers. Our direction is orthogonal to the Bayesian and ensemble approaches, where one can also combine them for even better performance.

**Contribution.** In this paper, we develop such a training method for detecting out-of-distribution $P_{\mathtt{out}}$ better without losing its original classification accuracy. First, we consider a new loss function, called *confidence loss*. Our key idea on the proposed loss is to additionally minimize the Kullback-Leibler (KL) divergence from the predictive distribution on out-of-distribution samples to the uniform one in order to give less confident predictions on them. Then, in- and out-of-distributions are expected to be more separable. However, optimizing the confidence loss requires training samples from out-of-distribution, which are often hard to sample: a priori knowledge on out-of-distribution is not available or its underlying space is too huge to cover. To handle the issue, we consider a new generative adversarial network (GAN) (Goodfellow et al., 2014) for generating most effective samples from $P_{\mathtt{out}}$. Unlike the original GAN, the proposed GAN generates 'boundary' samples in the low-density area of $P_{\mathtt{in}}$. Finally, we design a joint training scheme minimizing the classifier's loss and new GAN loss alternatively, i.e., the confident classifier improves the GAN, and vice versa, as training proceeds. Here, we emphasize that the proposed GAN does not need to generate explicit samples under our scheme, and instead it implicitly encourages training a more confident classifier.

We demonstrate the effectiveness of the proposed method using deep convolutional neural networks such as AlexNet (Krizhevsky, 2014) and VGGNet (Szegedy et al., 2015) for image classification tasks on CIFAR (Krizhevsky & Hinton, 2009), SVHN (Netzer et al., 2011), ImageNet (Deng et al., 2009), and LSUN (Yu et al., 2015) datasets. The classifier trained by our proposed method drastically improves the detection performance of all threshold-based detectors (Hendrycks & Gimpel, 2016; Liang et al., 2017) in all experiments. In particular, VGGNet with 13 layers trained by our method improves the true negative rate (TNR), i.e., the fraction of detected out-of-distribution (LSUN) samples, compared to the baseline: $14.0\% \rightarrow 39.1\%$ and $46.3\% \rightarrow 98.9\%$ on CIFAR-10 and SVHN, respectively, when 95% of in-distribution samples are correctly detected. We also provide visual understandings on the proposed method using the image datasets. We believe that our method can be a strong guideline when other researchers will pursue these tasks in the future.

## 2 TRAINING CONFIDENT NEURAL CLASSIFIERS

In this section, we propose a novel training method for classifiers in order to improve the performance of prior threshold-based detectors (Hendrycks & Gimpel, 2016; Liang et al., 2017) (see Appendix A for more details). Our motivation is that such inference algorithms can work better if the classifiers are trained so that they map the samples from in- and out-of-distributions into the output space separately. Namely, we primarily focus on training an improved classifier, and then use prior detectors under the trained model to measure its performance.

### 2.1 CONFIDENT CLASSIFIER FOR OUT-OF-DISTRIBUTION

Without loss of generality, suppose that the cross entropy loss is used for training. Then, we propose the following new loss function, termed confidence loss:

$$\min_{\theta} \ \mathbb{E}_{P_{\mathtt{in}}(\widehat{\mathbf{x}},\widehat{y})}\big[ -\log P_{\theta}\left(y = \widehat{y}|\widehat{\mathbf{x}}\right) \big] + \beta\mathbb{E}_{P_{\mathtt{out}}(\mathbf{x})}\big[ KL\left(\mathcal{U}\left(y\right) \parallel P_{\theta}\left(y|\mathbf{x}\right)\right) \big], \qquad (1)$$

where $KL$ denotes the Kullback-Leibler (KL) divergence, $\mathcal{U}\left(y\right)$ is the uniform distribution and $\beta > 0$ is a penalty parameter. It is highly intuitive as the new loss forces the predictive distribution

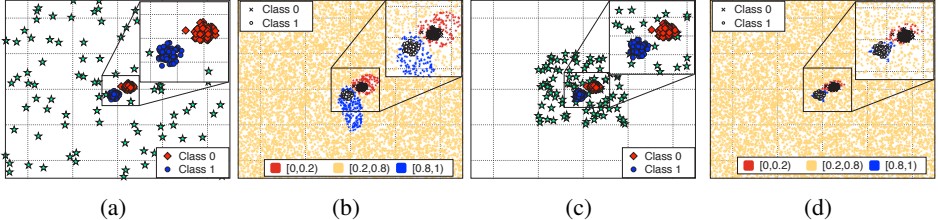

Figure 1: Illustrating the behavior of classifier under different out-of-distribution training datasets. We generate the out-of-distribution samples from (a) 2D box $[-50, 50]^2$, and show (b) the corresponding decision boundary of classifier. We also generate the out-of-distribution samples from (c) 2D box $[-20, 20]^2$, and show (d) the corresponding decision boundary of classifier.

on out-of-distribution samples to be closer to the uniform one, i.e., zero confidence, while that for samples from in-distribution still follows the label-dependent probability. In other words, the proposed loss is designed for assigning higher maximum prediction values, i.e., $\max_y P_\theta(y|\mathbf{x})$, to in-distribution samples than out-of-distribution ones. Here, a caveat is that adding the KL divergence term might degrade the classification performance. However, we found that it is not the case due to the high expressive power of deep neural networks, while in- and out-of-distributions become more separable with respect to the maximum prediction value by optimizing the confidence loss (see Section 3.1 for supporting experimental results).

We remark that minimizing a similar KL loss was studied recently for different purposes (Lee et al., 2017; Pereyra et al., 2017). Training samples for minimizing the KL divergence term is explicitly given in their settings while we might not. Ideally, one has to sample all (almost infinite) types of out-of-distribution to minimize the KL term in (1), or require some prior information on testing out-of-distribution for efficient sampling. However, this is often infeasible and fragile. To address the issue, we suggest to sample out-of-distribution close to in-distribution, which could be more effective in improving the detection performance, without any assumption on testing out-of-distribution.

In order to explain our intuition in details, we consider a binary classification task on a simple example, where each class data is drawn from a Gaussian distribution and entire data space is bounded by 2D box $[-50, 50]^2$ for visualization. We apply the confidence loss to simple fully-connected neural networks (2 hidden layers and 500 hidden units for each layer) using different types of out-of-distribution training samples. First, as shown in Figure 1(a), we construct an out-of-distribution training dataset of 100 (green) points using rejection sampling on the entire data space $[-50, 50]^2$. Figure 1(b) shows the decision boundary of classifier optimizing the confidence loss on the corresponding dataset. One can observe that a classifier still shows overconfident predictions (red and blue regions) near the labeled in-distribution region. On the other hand, if we construct a training out-of-distribution dataset of 100 points from $[-20, 20]^2$, i.e., closer to target, in-distribution space (see Figure 1(c)), a classifier produces confident predictions only on the labeled region and zero confidence on the remaining in the entire data space $[-50, 50]^2$ as shown in Figure 1(d). If one increases the number of training out-of-distribution samples which are generated from the entire space, i.e., $[-50, 50]^2$, Figure 1(b) is expected to be similar to Figure 1(d). In other words, one need more samples in order to train a confident classifier if samples are generated from the entire space. However, this might be impossible and not efficient since the number of out-of-distribution training samples might be almost infinite to cover its entire, huge actual data space. This implies that training out-of-distribution samples nearby the in-distribution region could be more effective in improving the detection performance. Our underlying intuition is that the effect of boundary of in-distribution region might propagate to the entire out-of-distribution space. Our experimental results in Section 3.1 also support this: realistic images are more useful as training out-of-distribution than synthetic datasets (e.g., Gaussian noise) for improving the detection performance when we consider an image classification task. This motivates us to develop a new generative adversarial network (GAN) for generating such effective out-of-distribution samples.

## 2.2 ADVERSARIAL GENERATOR FOR OUT-OF-DISTRIBUTION

In this section, we introduce a new training method for learning a generator of out-of-distribution inspired by generative adversarial network (GAN) (Goodfellow et al., 2014). We will first assume

that the classifier for in-distribution is fixed, and also describe the joint learning framework in the next section.

The GAN framework consists of two main components: discriminator $D$ and generator $G$. The generator maps a latent variable $\mathbf{z}$ from a prior distribution $P_{\text{pri}}(\mathbf{z})$ to generated outputs $G(\mathbf{z})$, and discriminator $D : \mathcal{X} \to [0,1]$ represents a probability that sample $\mathbf{x}$ is from a target distribution. Suppose that we want to recover the in-distribution $P_{\text{in}}(x)$ using the generator $G$. Then, one can optimize the following min-max objective for forcing $P_G \approx P_{\text{in}}$:

$$\min_G \max_D \ \mathbb{E}_{P_{\text{in}}(\mathbf{x})}\big[\log D(\mathbf{x})\big] + \mathbb{E}_{P_{\text{pri}}(\mathbf{z})}\big[\log(1 - D(G(\mathbf{z})))\big]. \tag{2}$$

However, unlike the original GAN, we want to make the generator recover an effective out-of-distribution $P_{\text{out}}$ instead of $P_{\text{in}}$. To this end, we propose the following new GAN loss:

$$\min_G \max_D \ \beta \underbrace{\mathbb{E}_{P_G(\mathbf{x})}\big[KL\left(\mathcal{U}(y) \parallel P_\theta(y|\mathbf{x})\right)\big]}_{(a)}$$

$$+ \underbrace{\mathbb{E}_{P_{\text{in}}(\mathbf{x})}\big[\log D(\mathbf{x})\big] + \mathbb{E}_{P_G(\mathbf{x})}\big[\log(1 - D(\mathbf{x}))\big]}_{(b)}, \tag{3}$$

where $\theta$ is the model parameter of a classifier trained on in-distribution. The above objective can be interpreted as follows: the first term (a) corresponds to a replacement of the out-of-distribution $P_{\text{out}}$ in (1)'s KL loss with the generator distribution $P_G$. One can note that this forces the generator to generate low-density samples since it can be interpreted as minimizing the log negative likelihood of in-distribution using the classifier, i.e., $P_{\text{in}}(\mathbf{x}) \approx \exp\left(KL\left(\mathcal{U}(y) \parallel P_\theta(y|\mathbf{x})\right)\right)$. We remark that this approximation is also closely related to the inception score (Salimans et al., 2016) which is popularly used as a quantitative measure of visual fidelity of the samples. The second term (b) corresponds to the original GAN loss since we would like to have out-of-distribution samples close to in-distribution, as mentioned in Section 2.1. Suppose that the model parameter of classifier $\theta$ is set appropriately such that the classifier produces the uniform distribution for out of distribution samples. Then, the KL divergence term (a) in (3) is approximately 0 no matter what out-of-distribution samples are generated. However, if the samples are far away from boundary, the GAN loss (b) in (3) should be high, i.e., the GAN loss forces having samples being not too far from the in-distribution space. Therefore, one can expect that proposed loss can encourage the generator to produce the samples which are on the low-density boundary of the in-distribution space. We also provide its experimental evidences in Section 3.2.

We also remark that (Dai et al., 2017) consider a similar GAN generating samples from out-of-distribution for the purpose of semi-supervised learning. The authors assume the existence of a pre-trained density estimation model such as PixelCNN++ (Salimans et al., 2017) for in-distribution, but such a model might not exist and be expensive to train in general. Instead, we use much simpler confident classifiers for approximating the density. Hence, under our fully-supervised setting, our GAN is much easier to train and more suitable.

## 2.3 Joint training method of confident classifier and adversarial generator

In the previous section, we suggest training the proposed GAN using a pre-trained confident classifier. We remind that the converse is also possible, i.e., the motivation of having such a GAN is for training a better classifier. Hence, two models can be used for improving each other. This naturally suggests a joint training scheme where the confident classifier improves the proposed GAN, and vice versa, as training proceeds. Specifically, we suggest the following joint objective function:

$$\min_G \max_D \min_\theta \ \underbrace{\mathbb{E}_{P_{\text{in}}(\widehat{\mathbf{x}},\widehat{y})}\big[-\log P_\theta(y = \widehat{y}|\widehat{\mathbf{x}})\big]}_{(c)} + \beta \underbrace{\mathbb{E}_{P_G(\mathbf{x})}\big[KL\left(\mathcal{U}(y) \parallel P_\theta(y|\mathbf{x})\right)\big]}_{(d)}$$

$$+ \underbrace{\mathbb{E}_{P_{\text{in}}(\widehat{\mathbf{x}})}\big[\log D(\widehat{\mathbf{x}})\big] + \mathbb{E}_{P_G(\mathbf{x})}\big[\log(1 - D(\mathbf{x}))\big]}_{(e)}. \tag{4}$$

The classifier's confidence loss corresponds to (c) + (d), and the proposed GAN loss corresponds to (d) + (e), i.e., they share the KL divergence term (d) under joint training. To optimize the above objective efficiently, we propose an alternating algorithm, which optimizes model parameters $\{\theta\}$ of classifier and GAN models $\{G, D\}$ alternatively as shown in Algorithm 1. Since the algorithm monotonically decreases the objective function, it is guaranteed to converge.

---

**Algorithm 1** Alternating minimization for detecting and generating out-of-distribution.

---

**repeat**

/∗ Update proposed GAN ∗/

Sample $\{\mathbf{z}_1, \ldots, \mathbf{z}_M\}$ and $\{\mathbf{x}_1, \ldots, \mathbf{x}_M\}$ from prior $P_{\mathtt{pri}}(\mathbf{z})$ and and in-distribution $P_{\mathtt{in}}(\mathbf{x})$, respectively, and update the discriminator $D$ by ascending its stochastic gradient of

$$\frac{1}{M} \sum_{i=1}^{M} \Big[ \log D\left(\mathbf{x}_i\right) + \log\left(1 - D\left(G\left(\mathbf{z}_i\right)\right)\right) \Big].$$

Sample $\{\mathbf{z}_1, \ldots, \mathbf{z}_M\}$ from prior $P_{\mathtt{pri}}(\mathbf{z})$, and update the generator $G$ by descending its stochastic gradient of

$$\frac{1}{M} \sum_{i=1}^{M} \Big[ \log\left(1 - D\left(G\left(\mathbf{z}_i\right)\right)\right) \Big] + \frac{\beta}{M} \sum_{i=1}^{M} \Big[ KL\left(\mathcal{U}\left(y\right) \parallel P_\theta\left(y|G\left(\mathbf{z}_i\right)\right)\right) \Big].$$

/∗ Update confident classifier ∗/

Sample $\{\mathbf{z}_1, \ldots, \mathbf{z}_M\}$ and $\{(\mathbf{x}_1, y_1), \ldots, (\mathbf{x}_M, y_M)\}$ from prior $P_{\mathtt{pri}}(\mathbf{z})$ and in-distribution $P_{\mathtt{in}}(\mathbf{x}, y)$, respectively, and update the classifier $\theta$ by descending its stochastic gradient of

$$\frac{1}{M} \sum_{i=1}^{M} \Big[ -\log P_\theta\left(y = y_i|\mathbf{x}_i\right) + \beta KL\left(\mathcal{U}\left(y\right) \parallel P_\theta\left(y|G\left(\mathbf{z}_i\right)\right)\right) \Big].$$

**until** *convergence*

---

## 3 EXPERIMENTAL RESULTS

We demonstrate the effectiveness of our proposed method using various datasets: CIFAR (Krizhevsky & Hinton, 2009), SVHN (Netzer et al., 2011), ImageNet (Deng et al., 2009), LSUN (Yu et al., 2015) and synthetic (Gaussian) noise distribution. We train convolutional neural networks (CNNs) including VGGNet (Szegedy et al., 2015) and AlexNet (Krizhevsky, 2014) for classifying CIFAR-10 and SVHN datasets. The corresponding test dataset is used as the in-distribution (positive) samples to measure the performance. We use realistic images and synthetic noises as the out-of-distribution (negative) samples. For evaluation, we measure the following metrics using the threshold-based detectors (Hendrycks & Gimpel, 2016; Liang et al., 2017): the true negative rate (TNR) at 95% true positive rate (TPR), the area under the receiver operating characteristic curve (AUROC), the area under the precision-recall curve (AUPR), and the detection accuracy, where larger values of all metrics indicate better detection performances. Due to the space limitation, more explanations about datasets, metrics and network architectures are given in Appendix B.[1]

| In-dist | Out-of-dist | Classification accuracy | TNR at TPR 95% | AUROC | Detection accuracy | AUPR in | AUPR out |
|---------|-------------|------------------------|----------------|-------|--------------------|---------|----------|
| | | | Cross entropy loss / Confidence loss | | | | |
| SVHN | CIFAR-10 (seen) | 93.82 / **94.23** | 47.4 / **99.9** | 62.6 / **99.9** | 78.6 / **99.9** | 71.6 / **99.9** | 91.2 / **99.4** |
| | TinyImageNet (unseen) | | 49.0 / **100.0** | 64.6 / **100.0** | 79.6 / **100.0** | 72.7 / **100.0** | 91.6 / **99.4** |
| | LSUN (unseen) | | 46.3 / **100.0** | 61.8 / **100.0** | 78.2 / **100.0** | 71.1 / **100.0** | 90.8 / **99.4** |
| | Gaussian (unseen) | | 56.1 / **100.0** | 72.0 / **100.0** | 83.4 / **100.0** | 77.2 / **100.0** | 92.8 / **99.4** |
| CIFAR-10 | SVHN (seen) | 80.14 / **80.56** | 13.7 / **99.8** | 46.6 / **99.9** | 66.6 / **99.8** | 61.4 / **99.9** | 73.5 / **99.8** |
| | TinyImageNet (unseen) | | **13.6** / 9.9 | **39.6** / 31.8 | **62.6** / 58.6 | **62.6** / 58.6 | **71.0** / 66.1 |
| | LSUN (unseen) | | **14.0** / 10.5 | **40.7** / 34.8 | **63.2** / 60.2 | **58.7** / 56.4 | **71.5** / 68.0 |
| | Gaussian (unseen) | | 2.8 / **3.3** | 10.2 / **14.1** | 50.0 / 50.0 | 48.1 / **49.4** | 39.9 / **47.0** |

Table 1: Performance of the baseline detector (Hendrycks & Gimpel, 2016) using VGGNet. All values are percentages and boldface values indicate relative the better results. For each in-distribution, we minimize the KL divergence term in (1) using training samples from an out-of-distribution dataset denoted by "seen", where other "unseen" out-of-distributions were only used for testing.

---

[1]Our code is available at `https://github.com/alinlab/Confident_classifier`.

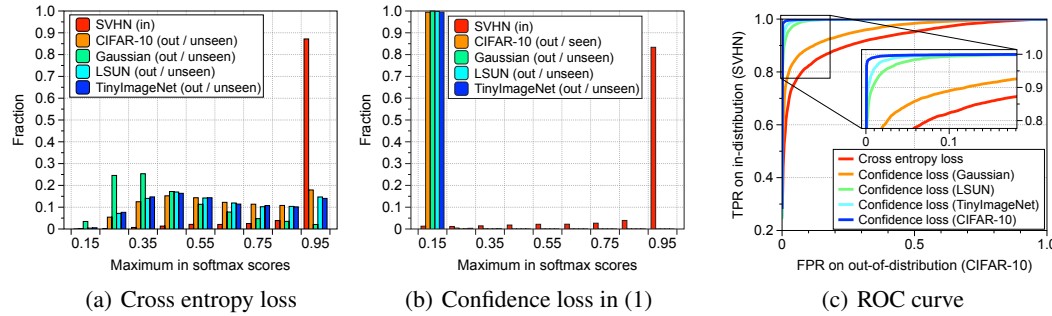

(a) Cross entropy loss      (b) Confidence loss in (1)      (c) ROC curve

Figure 2: For all experiments in (a), (b) and (c), we commonly use the SVHN dataset for in-distribution. Fraction of the maximum prediction value in softmax scores trained by (a) cross entropy loss and (b) confidence loss: the x-axis and y-axis represent the maximum prediction value and the fraction of images receiving the corresponding score, respectively. The receiver operating characteristic (ROC) curves under different losses are reported in (c): the red curve corresponds to the ROC curve of a model trained by optimizing the naive cross entropy loss, whereas other ones correspond to the ROC curves of models trained by optimizing the confidence loss. The KL divergence term in the confidence loss is optimized using explicit out-of-distribution datasets indicated in the parentheses, e.g., Confident loss (LSUN) means that we use the LSUN dataset for optimizing the KL divergence term.

## 3.1 EFFECTS OF CONFIDENCE LOSS

We first verify the effect of confidence loss in (1) trained by some explicit, say seen, out-of-distribution datasets. First, we compare the quality of confidence level by applying various training losses. Specifically, the softmax classifier is used and simple CNNs (two convolutional layers followed by three fully-connected layers) are trained by minimizing the standard cross entropy loss on SVHN dataset. We also apply the confidence loss to the models by additionally optimizing the KL divergence term using CIFAR-10 dataset (as training out-of-distribution). In Figure 2(a) and 2(b), we report distributions of the maximum prediction value in softmax scores to evaluate the separation quality between in-distribution (i.e., SVHN) and out-of-distributions. It is clear that there exists a better separation between the SVHN test set (red bar) and other ones when the model is trained by the confidence loss. Here, we emphasize that the maximum prediction value is also low on even untrained (unseen) out-of-distributions, e.g., TinyImageNet, LSUN and synthetic datasets. Therefore, it is expected that one can distinguish in- and out-of-distributions more easily when a classifier is trained by optimizing the confidence loss. To verify that, we obtain the ROC curve using the baseline detector (Hendrycks & Gimpel, 2016) that computes the maximum value of predictive distribution on a test sample and classifies it as positive (i.e., in-distribution) if the confidence score is above some threshold. Figure 2(c) shows the ROC curves when we optimize the KL divergence term on various datasets. One can observe that realistic images such as TinyImageNet (aqua line) and LSUN (green line) are more useful than synthetic datasets (orange line) for improving the detection performance. This supports our intuition that out-of-distribution samples close to in-distribution could be more effective in improving the detection performance as we discussed in Section 2.1.

We indeed evaluate the performance of the baseline detector for out-of-distribution using large-scale CNNs, i.e., VGGNets with 13 layers, under various training scenarios, where more results on AlexNet and ODIN detector (Liang et al., 2017) can be found in Appendix C (the overall trends of results are similar). For optimizing the confidence loss in (1), SVHN and CIFAR-10 training datasets are used for optimizing the KL divergence term for the cases when the in-distribution is CIFAR-10 and SVHN, respectively. Table 1 shows the detection performance for each in- and out-of-distribution pair. When the in-distribution is SVHN, the classifier trained by our method drastically improves the detection performance across all out-of-distributions without hurting its original classification performance. However, when the in-distribution is CIFAR-10, the confidence loss does not improve the detection performance in overall, where we expect that this is because the trained/seen SVHN out-of-distribution does not effectively cover all tested out-of-distributions. Our joint confidence loss in (4), which was designed under the intuition, resolves the issue of the CIFAR-10 (in-distribution) classification case in Table 1 (see Figure 4(b)).

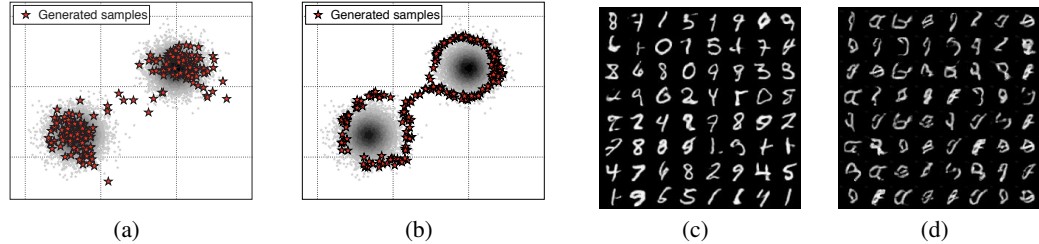

(a)           (b)           (c)           (d)

Figure 3: The generated samples from original GAN (a)/(c) and proposed GAN (b)/(d). In (a)/(b), the grey area is the 2D histogram of training in-distribution samples drawn from a mixture of two Gaussian distributions and red points indicate generated samples by GANs.

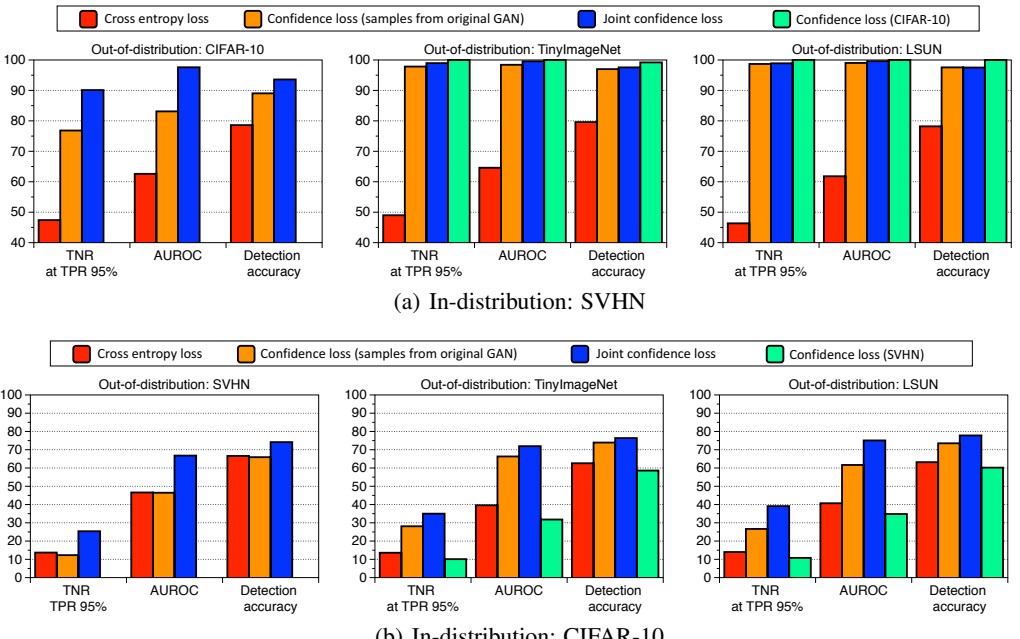

Figure 4: Performances of the baseline detector (Hendrycks & Gimpel, 2016) under various training losses. For training models by the confidence loss, the KL divergence term is optimized using samples indicated in the parentheses. For fair comparisons, we only plot the performances for unseen out-of-distributions, where those for seen out-of-distributions (used for minimizing the KL divergence term in (1)) can be found in Table 1.

## 3.2 EFFECTS OF ADVERSARIAL GENERATOR AND JOINT CONFIDENCE LOSS

In this section, we verify the effect of the proposed GAN in Section 2.2 and evaluate the detection performance of the joint confidence loss in (4). To verify that the proposed GAN can produce the samples nearby the low-density boundary of the in-distribution space, we first compare the generated samples by original GAN and proposed GAN on a simple example where the target distribution is a mixture of two Gaussian distributions. For both the generator and discriminator, we use fully-connected neural networks with 2 hidden layers. For our method, we use a pre-trained classifier which minimizes the cross entropy on target distribution samples and the KL divergence on out-of-distribution samples generated by rejection sampling on a bounded 2D box. As shown in Figure 3(a), the samples of original GAN cover the high-density area of the target distribution while those of proposed GAN does its boundary one (see Figure 3(b)). We also compare the generated samples of original and proposed GANs on MNIST dataset (LeCun et al., 1998), which consists of hand-written digits. For this experiment, we use deep convolutional GANs (DCGANs) (Radford et al., 2015). In this case, we use a pre-trained classifier which minimizes the cross entropy on MNIST

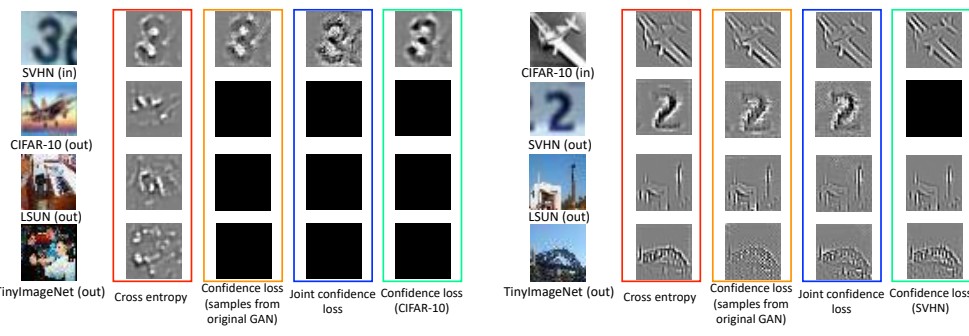

(a) In-distribution: SVHN        (b) In-distribution: CIFAR-10

Figure 5: Guided gradient (sensitivity) maps of the top-1 predicted class with respect to the input image under various training losses.

training samples and the KL divergence on synthetic Gaussian noises. As shown in Figure 3(c) and 3(d), samples of original GAN looks more like digits than those of proposed GAN. Somewhat interestingly, the proposed GAN still generates some new digit-like images.

We indeed evaluate the performance of our joint confidence loss in (4) utilizing the proposed GAN. To this end, we use VGGNets (as classifiers) and DCGANs (as GANs). We also test a variant of confidence loss which optimizes the KL divergence term on samples from a pre-trained original GAN (implicitly) modeling the in-distribution. One can expect that samples from the original GAN can be also useful for improving the detection performance since it may have bad generalization properties (Arora et al., 2017) and generate a few samples on the low-density boundary as like the proposed GAN. Figure 4 shows the performance of the baseline detector for each in- and out-of-distribution pair. First, observe that the joint confidence loss (blue bar) outperforms the confidence loss with some explicit out-of-distribution datasets (green bar). This is quite remarkable since the former is trained only using in-distribution datasets, while the latter utilizes additional out-of-distribution datasets. We also remark that our methods significantly outperform the baseline cross entropy loss (red bar) in all cases without harming its original classification performances (see Table 2 in Appendix C). Interestingly, the confidence loss with the original GAN (orange bar) is often (but not always) useful for improving the detection performance, whereas that with the proposed GAN (blue bar) still outperforms it in all cases.

Finally, we also provide visual interpretations of models using the guided gradient maps (Springenberg et al., 2014). Here, the gradient can be interpreted as an importance value of each pixel which influences on the classification decision. As shown in Figure 5, the model trained by the cross entropy loss shows sharp gradient maps for both samples from in- and out-of-distributions, whereas models trained by the confidence losses do only on samples from in-distribution. For the case of SVHN in-distribution, all confidence losses gave almost zero gradients, which matches to the results in Figure 4(a): their detection performances are almost perfect. For the case of CIFAR-10 distribution, one can now observe that there exists some connection between gradient maps and detection performances. This is intuitive because for detecting samples from out-of-distributions better, the classifier should look at more pixels as similar importance and the KL divergence term forces it. We think that our visualization results might give some ideas in future works for developing better inference methods for detecting out-of-distribution under our models.

## 4 CONCLUSION

In this paper, we aim to develop a training method for neural classification networks for detecting out-of-distribution better without losing its original classification accuracy. In essence, our method jointly trains two models for detecting and generating out-of-distribution by minimizing their losses alternatively. Although we primarily focus on image classification in our experiments, our method can be used for any classification tasks using deep neural networks. It is also interesting future directions applying our methods for other related tasks: regression (Malinin et al., 2017), network calibration (Guo et al., 2017), Bayesian probabilistic models (Li & Gal, 2017; Louizos & Welling, 2017), ensemble (Lakshminarayanan et al., 2017) and semi-supervised learning (Dai et al., 2017).

ACKNOWLEDGEMENTS

This work was supported in part by the Institute for Information & communications Technology Promotion(IITP) grant funded by the Korea government(MSIT) (No.2017-0-01778, Development of Explainable Human-level Deep Machine Learning Inference Framework), the ICT R&D program of MSIP/IITP [R-20161130-004520, Research on Adaptive Machine Learning Technology Development for Intelligent Autonomous Digital Companion], DARPA Explainable AI (XAI) program #313498 and Sloan Research Fellowship.

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

# A   THRESHOLD-BASED DETECTORS

In this section, we formally describe the detection procedure of threshold-based detectors (Hendrycks & Gimpel, 2016; Liang et al., 2017). For each data $\mathbf{x}$, it measures some confidence score $q(\mathbf{x})$ by feeding the data into a pre-trained classifier. Here, (Hendrycks & Gimpel, 2016) defined the confidence score as a maximum value of the predictive distribution, and (Liang et al., 2017) further improved the performance by processing the predictive distribution (see Appendix C.3 for more details). Then, the detector, $g(\mathbf{x}) : \mathcal{X} \to \{0, 1\}$, assigns label 1 if the confidence score $q(\mathbf{x})$ is above some threshold $\delta$, and label 0, otherwise:

$$g\left(\mathbf{x}\right) = \left\{ \begin{array}{cc} 1 & \text{if } q(\mathbf{x}) \geq \delta, \\ 0 & \text{otherwise.} \end{array} \right.$$

For this detector, we have to find a score threshold so that some positive examples are classified correctly, but this depends upon the trade-off between false negatives and false positives. To handle this issue, we use threshold-independent evaluation metrics such as area under the receiver operating characteristic curve (AUORC) and detection accuracy (see Appendix B).

# B   EXPERIMENTAL SETUPS IN SECTION 3

**Datasets.** We train deep models such as VGGNet (Szegedy et al., 2015) and AlexNet (Krizhevsky, 2014) for classifying CIFAR-10 and SVHN datasets: the former consists of 50,000 training and 10,000 test images with 10 image classes, and the latter consists of 73,257 training and 26,032 test images with 10 digits.[2] The corresponding test dataset are used as the in-distribution (positive) samples to measure the performance. We use realistic images and synthetic noises as the out-of-distribution (negative) samples: the TinyImageNet consists of 10,000 test images with 200 image classes from a subset of ImageNet images. The LSUN consists of 10,000 test images of 10 different scenes. We downsample each image of TinyImageNet and LSUN to size $32 \times 32$. The Gaussian noise is independently and identically sampled from a Gaussian distribution with mean $0.5$ and variance $1$. We clip each pixel value into the range $[0, 1]$.

**Detailed CNN structure and training.** The simple CNN that we use for evaluation shown in Figure 2 consists of two convolutional layers followed by three fully-connected layers. Convolutional layers have 128 and 256 filters, respectively. Each convolutional layer has a $5 \times 5$ receptive field applied with a stride of 1 pixel each followed by max pooling layer which pools $2 \times 2$ regions at strides of 2 pixels. AlexNet (Krizhevsky, 2014) consists of five convolutitonal layers followed by three fully-connected layers. Convolutional layers have 64, 192, 384, 256 and 256 filters, respectively. First and second convolutional layers have a $5 \times 5$ receptive field applied with a stride of 1 pixel each followed by max pooling layer which pools $3 \times 3$ regions at strides of 2 pixels. Other convolutional layers have a $3 \times 3$ receptive field applied with a stride of 1 pixel followed by max pooling layer which pools $2 \times 2$ regions at strides of 2 pixels. Fully-connected layers have 2048, 1024 and 10 hidden units, respectively. Dropout (Hinton et al., 2012) was applied to only fully-connected layers of the network with the probability of retaining the unit being 0.5. All hidden units are ReLUs. Figure 6 shows the detailed structure of VGGNet (Szegedy et al., 2015) with three fully-connected layers and 10 convolutional layers. Each ConvReLU box in the figure indicates a $3 \times 3$ convolutional layer followed by ReLU activation. Also, all max pooling layers have $2 \times 2$ receptive fields with stride 2. Dropout was applied to only fully-connected layers of the network with the probability of retaining the unit being 0.5. For all experiments, the softmax classifier is used, and each model is trained by optimizing the objective function using Adam learning rule (Kingma & Ba, 2014). For each out-of-distribution dataset, we randomly select 1,000 images for tuning the penalty parameter $\beta$, mini-batch size and learning rate. The penalty parameter is chosen from $\beta \in \{0, 0.1, \dots 1.9, 2\}$, the mini-batch size is chosen from $\{64, 128\}$ and the learning rate is chosen from $\{0.001, 0.0005, 0.0002\}$. The optimal parameters are chosen to minimize the detection error on the validation set. We drop the learning rate by 0.1 at 60 epoch and models are trained for total 100 epochs. The best test result is reported for each method.

**Performance metrics.** We measure the following metrics using threshold-based detectors:

---

[2]We do not use the extra SVHN dataset for training.

- **True negative rate (TNR) at 95% true positive rate (TPR).** Let TP, TN, FP, and FN denote true positive, true negative, false positive and false negative, respectively. We measure TNR = TN / (FP+TN), when TPR = TP / (TP+FN) is 95%.

- **Area under the receiver operating characteristic curve (AUROC).** The ROC curve is a graph plotting TPR against the false positive rate = FP / (FP+TN) by varying a threshold.

- **Area under the precision-recall curve (AUPR).** The PR curve is a graph plotting the precision = TP / (TP+FP) against recall = TP / (TP+FN) by varying a threshold. AUPR-IN (or -OUT) is AUPR where in- (or out-of-) distribution samples are specified as positive.

- **Detection accuracy.** This metric corresponds to the maximum classification probability over all possible thresholds $\delta$: $1 - \min_\delta \left\{ P_{\text{in}}\left(q\left(\mathbf{x}\right) \leq \delta\right) P\left(\mathbf{x} \text{ is from } P_{\text{in}}\right) + P_{\text{out}}\left(q\left(\mathbf{x}\right) > \delta\right) P\left(\mathbf{x} \text{ is from } P_{\text{out}}\right) \right\}$, where $q(\mathbf{x})$ is a confident score such as a maximum value of softmax. We assume that both positive and negative examples have equal probability of appearing in the test set, i.e., $P\left(\mathbf{x} \text{ is from } P_{\text{in}}\right) = P\left(\mathbf{x} \text{ is from } P_{\text{out}}\right) = 0.5$.

Note that AUROC, AUPR and detection accuracy are threshold-independent evaluation metrics.

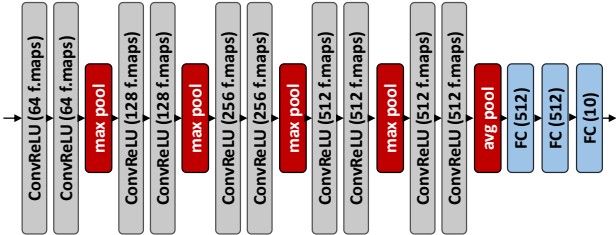

Figure 6: Detailed structure of VGGNet with 13 layers.

**Generating samples on a simple example.** As shown in Figure 3(a) and Figure 3(b), we compare the generated samples by original GAN and proposed GAN on a simple example where the target distribution is a mixture of two Gaussian distributions. For both the generator and discriminator, we use fully-connected neural networks with 2 hidden layers and 500 hidden units for each layer. For all layers, we use ReLU activation function. We use a 100-dimensional Gaussian prior for the latent variable $\mathbf{z}$. For our method, we pre-train the simple fully-connected neural networks (2 hidden layers and 500 ReLU units for each layer) by minimizing the cross entropy on target distribution samples and the KL divergence on out-of-distribution samples generated by rejection sampling on bounded 2D box $[-10, 10]^2$. The penalty parameter $\beta$ is set to 1. We use ADAM learning rule (Kingma & Ba, 2014) with a mini-batch size of 400. The initial learning rate is set to 0.002, and we train for total 100 epochs.

**Generating samples on MNIST.** As shown in Figure 3(c) and Figure 3(d), we compare the generated samples of original and proposed GANs on MNIST dataset, which consists of greyscale images, each containing a digit 0 to 9 with 60,000 training and 10,000 test images. We expand each image to size $3 \times 32 \times 32$. For both the generator and discriminator, we use deep convolutional GANs (DCGANs) (Radford et al., 2015). The discriminator and generator consist of four convolutional and deconvolutional layers, respectively. Convolutional layers have 128, 256, 512 and 1 filters, respectively. Each convolutional layer has a $4 \times 4$ receptive field applied with a stride of 2 pixel. The second and third convolutional layers are followed by batch normalization (Ioffe & Szegedy, 2015). For all layers, we use LeakyReLU activation function. Deconvolutional layers have 512, 256, 128 and 1 filters, respectively. Each deconvolutional layer has a $4 \times 4$ receptive field applied with a stride of 2 pixel followed by batch normalization (Ioffe & Szegedy, 2015) and ReLU activation. For our method, we use a pre-trained simple CNNs (two convolutional layers followed by three fully-connected layers) by minimizing the cross entropy on MNIST training samples and the KL divergence on synthetic Gaussian noise. Convolutional layers have 128 and 256 filters, respectively. Each convolutional layer has a $5 \times 5$ receptive field applied with a stride of 1 pixel each followed by max pooling layer which pools $2 \times 2$ regions at strides of 2 pixels. The penalty parameter $\beta$ is set to 1. We use ADAM learning rule (Kingma & Ba, 2014) with a mini-batch size of 128. The initial learning rate is set to 0.0002, and we train for total 50 epochs.

## C    More experimental results

### C.1    Classification performances

Table 2 reports the classification accuracy of VGGNets on CIFAR-10 and SVHN datasets under various training losses shown in Figure 4. One can note that all methods do not degrade the original classification performance, where the differences in classification errors of across all tested single models are at most $1\%$ in our experiments.

| In-distribution | Cross entropy | Confidence loss with original GAN | Joint confidence loss | Confidence loss with explicit out-of-distribution samples |
|---|---|---|---|---|
| CIFAR-10 | 80.14 | 80.27 | 81.39 | 80.56 |
| SVHN | 93.82 | 94.08 | 93.81 | 94.23 |

Table 2: Classification test set accuracy of VGGNets on CIFAR-10 and SVHN datasets under various training losses.

### C.2    Calibration effects of confidence loss

We also verify the calibration effects (Guo et al., 2017) of our methods: whether a classifier trained by our method can indicate when they are likely to be incorrect for test samples from the in-distribution. In order to evaluate the calibration effects, we measure the expected calibration error (ECE) (Naeini et al., 2015). Given test data $\{(x_1, y_1), \ldots, (x_n, y_n)\}$, we group the predictions into $M$ interval bins (each of size $1/M$). Let $B_m$ be the set of indices of samples whose prediction confidence falls into the interval $\left(\frac{m-1}{M}, \frac{m}{M}\right]$. Then, the accuracy of $B_m$ is defined as follows:

$$\text{acc}(B_m) = \frac{1}{|B_m|} \sum_{i \in B_m} 1_{\{y_i = \arg\max_y P_\theta(y|\mathbf{x}_i)\}},$$

where $\theta$ is the model parameters of a classifier and $1_A \in \{0, 1\}$ is the indicator function for event $A$. We also define the confidence of $B_m$ as follows:

$$\text{conf}(B_m) = \frac{1}{|B_m|} \sum_{i \in B_m} q(\mathbf{x}_i),$$

where $q(\mathbf{x}_i)$ is the confidence of data $i$. Using these notations, we measure the expected calibration error (ECE) as follows:

$$\text{ECE} = \sum_{m=1}^{M} \frac{|B_m|}{n} |\text{acc}(B_m) - \text{conf}(B_m)|.$$

One can note that ECE is zero if the confidence score can represent the true distribution. Table 3 shows the calibration effects of confidence loss when we define the confidence score $q$ as the maximum predictive distribution of a classifier. We found that the ECE of a classifier trained by our methods is lower than that of a classifier trained by the standard cross entropy loss. This implies that our proposed method is effective at calibrating predictions. We also remark that the temperature scaling (Guo et al., 2017) provides further improvements under a classifier trained by our joint confidence loss.

| In-distribution | Without temperature scaling | | With temperature scaling | |
|---|---|---|---|---|
| | Cross entropy loss | Joint confidence loss | Cross entropy loss | Joint confidence loss |
| CIFAR-10 | 18.45 | 14.62 | 7.07 | **6.19** |
| SVHN | 5.30 | 5.13 | 2.80 | **1.39** |

Table 3: Expected calibration error (ECE) of VGGNets on CIFAR-10 and SVHN datasets under various training losses. The number of bins $M$ is set to 20. All values are percentages and boldface values indicate relative the better results.

## C.3 EXPERIMENTAL RESULTS USING ODIN DETECTOR

In this section, we verify the effects of confidence loss using ODIN detector (Liang et al., 2017) which is an advanced threshold-based detector using temperature scaling (Guo et al., 2017) and input perturbation. The key idea of ODIN is the temperature scaling which is defined as follows:

$$P_\theta(y = \widehat{y}|\mathbf{x}; T) = \frac{\exp\left(f_{\widehat{y}}(\mathbf{x})/T\right)}{\sum_y \exp\left(f_y(\mathbf{x})/T\right)},$$

where $T > 0$ is the temperature scaling parameter and $\mathbf{f} = (f_1, \ldots, f_K)$ is final feature vector of neural networks. For each data $\mathbf{x}$, ODIN first calculates the pre-processed image $\widehat{\mathbf{x}}$ by adding the small perturbations as follows:

$$\mathbf{x}' = \mathbf{x} - \varepsilon \mathrm{sign}\left(-\bigtriangledown_\mathbf{x} \log P_\theta(y = \widehat{y}|\mathbf{x}; T)\right),$$

where $\varepsilon$ is a magnitude of noise and $\widehat{y}$ is the predicted label. Next, ODIN feeds the pre-processed data into the classifier, computes the maximum value of scaled predictive distribution, i.e., $\max_y P_\theta(y|\mathbf{x}'; T)$, and classifies it as positive (i.e., in-distribution) if the confidence score is above some threshold $\delta$.

For ODIN detector, the perturbation noise $\varepsilon$ is chosen from $\{0, 0.0001, 0.001, 0.01\}$, and the temperature $T$ is chosen from $\{1, 10, 100, 500, 1000\}$. The optimal parameters are chosen to minimize the detection error on the validation set. Figure 7 shows the performance of the OIDN and baseline detector for each in- and out-of-distribution pair. First, we remark that the baseline detector using classifiers trained by our joint confidence loss (blue bar) typically outperforms the ODIN detector using classifiers trained by the cross entropy loss (orange bar). This means that our classifier can map in- and out-of-distributions more separately without pre-processing methods such as temperature scaling. The ODIN detector provides further improvements if one uses it with our joint confidence loss (green bar). In other words, our proposed training method can improve all prior detection methods.

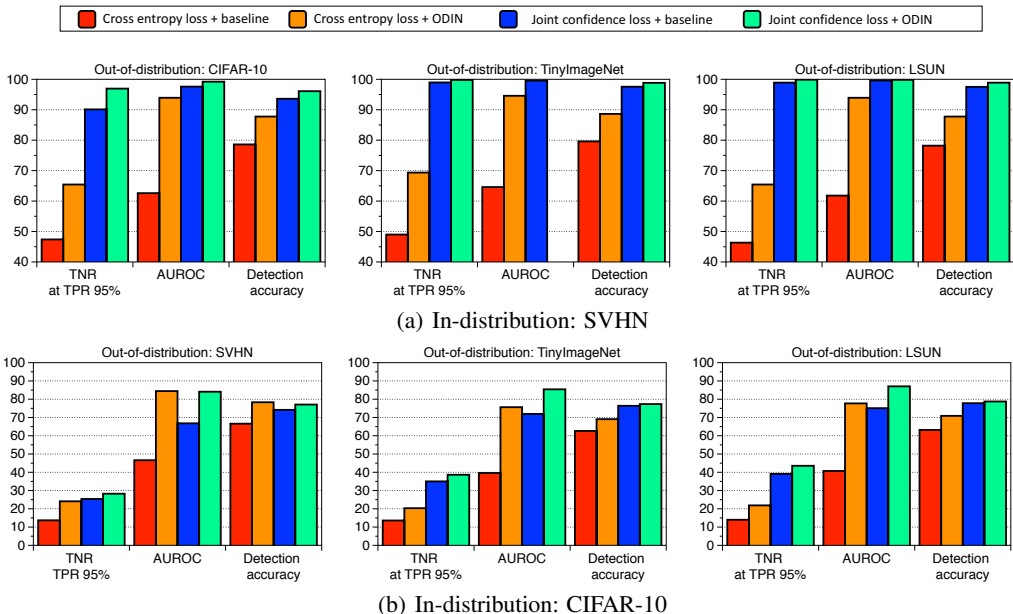

Figure 7: Performances of the baseline detector (Hendrycks & Gimpel, 2016) and ODIN detector (Liang et al., 2017) under various training losses.

## C.4 EXPERIMENTAL RESULTS ON ALEXNET

Table 4 shows the detection performance for each in- and out-of-distribution pair when the classifier is AlexNet (Krizhevsky, 2014), which is one of popular CNN architectures. We remark that they show similar trends.

| In-dist | Out-of-dist | Classification accuracy | TNR at TPR 95% | AUROC | Detection accuracy | AUPR in | AUPR out |
|---|---|---|---|---|---|---|---|
| | | | Cross entropy loss / Confidence loss | | | | |
| Baseline (SVHN) | CIFAR-10 (seen) | 92.14 / **93.77** | 42.0 / **99.9** | 88.0 / **100.0** | 83.4 / **99.8** | 88.7 / **99.9** | 87.3 / **99.3** |
| | TinyImageNet (unseen) | | 45.6 / **99.9** | 89.4 / **100.0** | 84.3 / **99.9** | 90.2 / **100.0** | 88.6 / **99.3** |
| | LSUN (unseen) | | 44.6 / **100.0** | 89.8 / **100.0** | 84.5 / **99.9** | 90.8 / **100.0** | 88.4 / **99.3** |
| | Gaussian (unseen) | | 58.6 / **100.0** | 94.2 / **100.0** | 88.8 / **100.0** | 95.5 / **100.0** | 92.5 / **99.3** |
| Baseline (CIFAR-10) | SVHN (seen) | **76.58** / 76.18 | 12.8 / **99.6** | 71.0 / **99.9** | 73.2 / **99.6** | 74.3 / **99.9** | 70.7 / **99.6** |
| | TinyImageNet (unseen) | | **10.3** / 10.1 | **59.2** / 52.1 | **64.2** / 62.0 | **63.6** / 59.8 | **64.4** / 62.3 |
| | LSUN (unseen) | | **10.7** / 8.1 | **56.3** / 51.5 | **64.3** / 61.8 | **62.3** / 59.5 | **65.3** / 61.6 |
| | Gaussian (unseen) | | **6.7** / 1.0 | **49.6** / 13.5 | **61.3** / 50.0 | **58.5** / 43.7 | **59.5** / 32.0 |
| ODIN (SVHN) | CIFAR-10 (seen) | 92.14 / **93.77** | 55.5 / **99.9** | 89.1 / **99.2** | 82.4 / **99.8** | 85.9 / **100.0** | 89.0 / **99.2** |
| | TinyImageNet (unseen) | | 59.5 / **99.9** | 90.5 / **99.3** | 83.8 / **99.9** | 87.5 / **100.0** | 90.4 / **99.3** |
| | LSUN (unseen) | | 61.5 / **100.0** | 91.8 / **99.3** | 84.8 / **99.9** | 90.5 / **100.0** | 91.3 / **99.3** |
| | Gaussian (unseen) | | 82.6 / **100.0** | 97.0 / **99.3** | 91.6 / **100.0** | 97.4 / **100.0** | 96.4 / **99.3** |
| ODIN (CIFAR-10) | SVHN (seen) | **76.58** / 76.18 | 37.1 / **99.6** | 86.7 / **99.6** | 79.3 / **99.6** | 88.1 / **99.9** | 84.2 / **99.6** |
| | TinyImageNet (unseen) | | **11.4** / 8.4 | **69.1** / 65.6 | **64.4** / 61.8 | **71.4** / 68.6 | **64.6** / 60.7 |
| | LSUN (unseen) | | **13.3** / 7.1 | **71.9** / 67.1 | **75.3** / 63.7 | **67.2** / 72.0 | **65.3** / 60.5 |
| | Gaussian (unseen) | | **3.8** / 0.0 | **70.9** / 57.2 | **69.3** / 40.4 | **78.1** / 56.1 | **60.7** / 40.7 |

Table 4: Performance of the baseline detector (Hendrycks & Gimpel, 2016) and ODIN detector (Liang et al., 2017) using AlexNet. All values are percentages and boldface values indicate relative the better results. For each in-distribution, we minimize the KL divergence term in (1) using training samples from an out-of-distribution dataset denoted by "seen", where other "unseen" out-of-distributions were also used for testing.

## D MAXIMIZING ENTROPY

One might expect that the entropy of out-of-distribution is expected to be much higher compared to that of in-distribution since the out-of-distribution is typically on a much larger space than the in-distribution. Therefore, one can add maximizing the entropy of generator distribution to new GAN loss in (3) and joint confidence loss in (4). However, maximizing the entropy of generator distribution is technically challenging since a GAN does not model the generator distribution explicitly. To handle the issue, one can leverage the pull-away term (PT) (Zhao et al., 2017):

$$-\mathcal{H}\left(P_G\left(\mathbf{x}\right)\right) \simeq \mathcal{PT}\left(P_G\left(\mathbf{x}\right)\right) = \frac{1}{M(M-1)} \sum_{i=1}^{M} \sum_{j \neq i} \left( \frac{G\left(\mathbf{z}_i\right)^\top G\left(\mathbf{z}_j\right)}{\|G\left(\mathbf{z}_i\right)\|\|G\left(\mathbf{z}_j\right)\|} \right)^2,$$

where $\mathcal{H}\left(\cdot\right)$ denotes the entropy, $\mathbf{z}_i, \mathbf{z}_j \sim P_{\texttt{pri}}\left(\mathbf{z}\right)$ and $M$ is the number of samples. Intuitively, one can expect the effect of increasing the entropy by minimizing PT since it corresponds to the squared cosine similarity of generated samples. We note that (Dai et al., 2017) also used PT to maximize the entropy. Similarly as in Section 3.2, we verify the effects of PT using VGGNet. Table 5 shows the performance of the baseline detector for each in- and out-of-distribution pair. We found that joint confidence loss with PT tends to (but not always) improve the detection performance. However, since PT increases the training complexity and the gains from PT are relatively marginal (or controversial), we leave it as an auxiliary option for improving the performance.

| In-dist | Out-of-dist | Classification accuracy | TNR at TPR 95% | AUROC | Detection accuracy | AUPR in | AUPR out |
|---|---|---|---|---|---|---|---|
| | | | Joint confidence loss without PT / with PT | | | | |
| SVHN | CIFAR-10 | 93.81 / **94.05** | 90.1 / **92.3** | 97.6 / **98.1** | 93.6 / **94.6** | 97.7 / **98.2** | 97.9 / **98.7** |
| | TinyImageNet | | 99.0 / **99.9** | 99.6 / **100.0** | 97.6 / **99.7** | 99.7 / **100.0** | 94.5 / **100.0** |
| | LSUN | | 98.9 / **100.0** | 99.6 / **100.0** | 97.5 / **99.9** | 99.7 / **100.0** | 95.5 / **100.0** |
| CIFAR-10 | SVHN | **81.39** / 80.60 | **25.4** / 13.2 | 66.8 / **69.5** | 74.2 / **75.1** | 71.3 / **73.5** | **78.3** / 72.0 |
| | TinyImageNet | | 35.0 / **44.8** | 72.0 / **78.4** | 76.4 / **77.6** | 74.7 / **79.4** | 82.2 / **84.4** |
| | LSUN | | 39.1 / **49.1** | 75.1 / **80.7** | 77.8 / **78.7** | 77.1 / **81.3** | 83.6 / **85.8** |

Table 5: Performance of the baseline detector (Hendrycks & Gimpel, 2016) using VGGNets trained by joint confidence loss with and without pull-away term (PT). All values are percentages and boldface values indicate relative the better results.

# E    ADDING OUT-OF-DISTRIBUTION CLASS

Instead of forcing the predictive distribution on out-of-distribution samples to be closer to the uniform one, one can simply add an additional "out-of-distribution" class to a classifier as follows:

$$\min_{\theta} \ \mathbb{E}_{P_{\text{in}}(\widehat{\mathbf{x}},\widehat{y})}\big[ -\log P_{\theta}\left(y = \widehat{y}|\widehat{\mathbf{x}}\right) \big] + \mathbb{E}_{P_{\text{out}}(\mathbf{x})}\big[ -\log P_{\theta}\left(y = K + 1|\mathbf{x}\right) \big], \qquad (5)$$

where $\theta$ is a model parameter. Similarly as in Section 3.1, we compare the performance of the confidence loss with that of above loss in (5) using VGGNets with for image classification on SVHN dataset. To optimize the KL divergence term in confidence loss and the second term of (5), CIFAR-10 training datasets are used. In order to compare the detection performance, we define the the confidence score of input $\mathbf{x}$ as $1 - P_{\theta}\left(y = K + 1|\mathbf{x}\right)$ in case of (5). Table 6 shows the detection performance for out-of-distribution. First, the classifier trained by our method often significantly outperforms the alternative adding the new class label. This is because modeling explicitly out-of-distribution can incur overfitting to trained out-of-distribution dataset.

| Detector | Out-of-dist | Classification accuracy | TNR at TPR 95% | AUROC | Detection accuracy | AUPR in | AUPR out |
|---|---|---|---|---|---|---|---|
| | | | $K + 1$ class loss in (5) / Confidence loss | | | | |
| Baseline detector | SVHN (seen) | 79.61 / **80.56** | 99.6 / **99.8** | 99.8 / **99.9** | 99.7 / **99.8** | 99.8 / **99.9** | **99.9** / 99.8 |
| | TinyImageNet (unseen) | | 0.0 / **9.9** | 5.2 / **31.8** | 51.3 / **58.6** | 50.7 / **55.3** | 64.3 / **66.1** |
| | LSUN (unseen) | | 0.0 / **10.5** | 5.6 / **34.8** | 51.5 / **60.2** | 50.8 / **56.4** | **71.5** / 68.0 |
| | Gaussian (unseen) | | 0.0 / **3.3** | 0.1 / **14.1** | 50.0 / 50.0 | 49.3 / **49.4** | 12.2 / **47.0** |
| ODIN detector | SVHN (seen) | 79.61 / **80.56** | 99.6 / **99.8** | **99.9** / 99.8 | 99.1 / **99.8** | 99.9 / 99.9 | **99.9** / 99.8 |
| | TinyImageNet (unseen) | | 0.3 / **12.2** | 47.3 / **70.6** | 55.12 / **65.7** | 56.6 / **72.7** | 44.3 / **65.6** |
| | LSUN (unseen) | | 0.1 / **13.7** | 48.3 / **73.1** | 55.9 / **67.9** | 57.5 / **75.2** | 44.7 / **67.8** |
| | Gaussian (unseen) | | 0.0 / **8.2** | 28.3 / **68.3** | 54.4 / **65.4** | 47.8 / **74.1** | 36.8 / **61.5** |

Table 6: Performance of the baseline detector (Hendrycks & Gimpel, 2016) and ODIN detector (Liang et al., 2017) using VGGNet. All values are percentages and boldface values indicate relative the better results.

