# OpenReview forum: "Training Confidence-calibrated Classifiers for Detecting Out-of-Distribution Samples"
_ICLR.cc/2018/Conference — Accept (Poster)_

### Official Review · AnonReviewer2 · 2017-11-27
**Interesting idea, but not yet convinced**

**Rating:** 6
**Confidence:** 4

**Review:**

I have read authors' reply.  In response to authors' comprehensive reply and feedback. I upgrade my score to 6.

-----------------------------

This paper presents a novel approach to calibrate classifiers for out of distribution samples. In additional to the original cross entropy loss, the “confidence loss”  was proposed to guarantee the out of distribution points have low confidence in the classifier. As out of distribution samples are hard to obtain, authors also propose to use GAN generating “boundary” samples as out of distribution samples.

The problem setting is new and objective (1) is interesting and reasonable. However, I am not very convinced that objective (3) will generate boundary samples. Suppose that theta is set appropriately so that p_theta (y|x) gives a uniform distribution over labels for out of distribution samples. Because of the construction of U(y), which uniformly assign labels to generated out of distribution samples, the conditional probability p_g (y|x) should always be uniform so p_g (y|x) divided by p_theta (y|x) is almost always 1. The KL divergence in (a) of (3) should always be approximately 0 no matter what samples are generated.

I also have a few other concerns:
1. There seems to be a related work:
[1] Perello-Nieto et al., Background Check: A general technique to build more reliable and versatile classifiers, ICDM 2016,
Where authors constructed a classifier, which output K+1 labels and the K+1-th label is the “background noise” label for this classification problem. Is the method in [1] applicable to this paper’s setting?  Moreover, [1] did not seem to generate any out of distribution samples.

2. I am not so sure that how the actual out of distribution detection was done (did I miss something here?). Authors repeatedly mentioned “maximum prediction values”, but it was not defined throughout the paper.
Algorithm 1. is called “minimization for detection and generating out of distribution (samples)”, but this is only gradient descent, right? I do not see a detection procedure. Given the title also contains “detecting”, I feel authors should write explicitly how the detection is done in the main body.

---

> ### Author Response · Authors · 2017-12-18
> **Our responses for AnonReviewer2**
>
> We very much appreciate your valuable comments, efforts and times on our paper. We provide our responses for all questions below. Revised parts in the new draft are colored by blue.
>
> Q1: "I am not very convinced that objective (3) will generate boundary samples."
>
> A1: As you pointed out, the KL divergence term (a) of (3) is approximately 0 no matter how out-of-distribution samples are generated. However, if the samples are far away from "boundary" (here, we assume the high-density area of in-distribution is a closed set), the GAN loss (b) of (3) should be high, i.e., the GAN loss forces having samples being not too far from the in-distribution space. This is the primary reason why (3) will generate out-of-distribution samples from the low-density boundary of the in-distribution space. We provided more explanations in the revision (please see updated Section 2.2 for details).
>
> Q2: "There seems to be a related work:
> [1] Perello-Nieto et al., Background Check: A general technique to build more reliable and versatile classifiers, ICDM 2016, Where authors constructed a classifier, which output K+1 labels and the K+1-th label is the "background noise" label for this classification problem. Is the method in [1] applicable to this paper's setting? Moreover, [1] did not seem to generate any out of distribution samples."
>
> A2: As you mentioned, the Background Check (BC) proposed by [1] can be applied to our setting, i.e., one can consider background distribution in [1] as out-of-distribution. The authors propose two methods called discriminative approach (BCD) and familiarity approach (BCF). First, BCD requires out-of-distribution samples, but they mentioned generating artificial background data is hard and did not try in their experiments. This is what we resolve in this paper. On the other hand, BCF does not require out-of-distribution samples, and instead it uses a density estimator of P_{in}(x) such as one-class support vector machine (OCSVM) for modeling in-distributions. Such an additional model not only increases the detection complexity, but also is not clear to perform well in the high-dimensional datasets used in our paper. For example, one can try complex density estimators such as PixelCNN [2], but they are quite difficult to train and should be chosen depending on data characteristics (this hurts the generality of our work). Our method is much simpler and easier to use. In addition, please note that the authors [1] did not report any neural network experiments at all.
>
> Q3: "Authors repeatedly mentioned "maximum prediction values", but it was not defined throughout the paper."
>
> A3: The "maximum prediction value" corresponds to a maximum value of the predictive distribution, i.e., \max_y P(y|x). We formally defined this in the revision (Section 2.1).
>
> Q4: "I am not so sure that how the actual out of distribution detection was done (did I miss something here?). Algorithm 1. is called "minimization for detection and generating out of distribution (samples)", but this is only gradient descent, right? I do not see a detection procedure. Given the title also contains "detecting", I feel authors should write explicitly how the detection is done in the main body."
>
> A4: Our goal is to develop a new training method (Algorithm 1 is the training algorithm), which works with a simple detection method. For the actual detecting procedure, one can apply any known inference algorithms [3,4,5] on a trained model. Here, we remark that the performance of these detectors highly depends how the classifier is trained, e.g., as shown in Table 1 and Figure 4, the detection performance of prior inference algorithms can be dramatically improved under a confident classifier trained by our method. We explained the detection procedure more precisely in the revision, as mentioned in the beginning of Section 2 and more formally defined in the Appendix A. Thank you for your suggestion.
>
> [1] Perello-Nieto et al., Background Check: A general technique to build more reliable and versatile classifiers, ICDM 2016
> [2] Oord, A.V.D., Kalchbrenner, N., Vinyals, O., Espeholt, L., Graves, A. and Kavukcuoglu, K. Conditional Image Generation with PixelCNN Decoders. In NIPS, 2016.
> [3] Shiyu Liang, Yixuan Li, and R Srikant. Principled detection of out-of-distribution examples in neural networks. arXiv preprint arXiv:1706.02690, 2017.
> [4] Chuan Guo, Geoff Pleiss, Yu Sun, and Kilian Q Weinberger. On calibration of modern neural networks. In ICML, 2017.
> [5] Dan Hendrycks and Kevin Gimpel. A baseline for detecting misclassified and out-of-distribution examples in neural networks. In ICLR, 2017.
>
> Thanks,
> Authors.

---

### Official Review · AnonReviewer3 · 2017-11-27
**interesting idea for robust classification**

**Rating:** 7
**Confidence:** 3

**Review:**

The manuscript proposes a generative approach to detect which samples are within vs. out of the sample space of the training distribution. This distribution is used to adjust the classifier so it makes confident predictions within sample, and less confident predictions out of sample, where presumably it is prone to mistakes. Evaluation on several datasets suggests that accounting for the within-sample distribution in this way can often actually improve evaluation performance, and can help the model detect outliers.

The manuscript is reasonably well written overall, though some of the writing could be improved e.g. a clearer description of the cost function in section 2. However, equation 4 and algorithm 1 were very helpful in clarifying the cost function. The manuscript also does a good job giving pointers to related prior work. The problem of interest is timely and important, and the provided solution seems reasonable and is well evaluated.

Looking at the cost function and the intuition, the difference in figure 1 seems to be primarily due to the relative number of samples used during optimization -- and not to anything inherent about the distribution as is claimed. In particular, if a proportional number of samples is generated for the 50x50 case, I would expect the plots to be similar. I suggest the authors modify the claim of figure 1 accordingly.

Along those lines, it would be interesting if instead of the uniform distribution, a model that explicitly models within vs. out of sample might perform better? Though this is partially canceled out by the other terms in the optimization.

Finally, the authors claim that the PT is approximately equal to entropy. The cited reference (Zhao et. al. 2017) does not justify the claim. I suggest the authors remove this claim or correctly justify it.

Questions:
 - Could the authors comment on cases where such a strong within-sample assumption may adversely affect performance?
 - Could the authors comment on how the modifications affect prediction score calibration?
 - Could the authors comment on whether they think the proposed approach may be more resilient to adversarial attacks?

Minor issues:
 - Figure 1 is unclear using dots. Perhaps the authors can try plotting a smoothed decision boundary to clarify the idea?

---

> ### Author Response · Authors · 2017-12-18
> **Our responses for AnonReviewer 3**
>
> We very much appreciate your valuable comments, efforts and times on our paper. We provide our responses for all questions below. Revised parts in the new draft are colored by blue.
>
> Q1: "About the difference in figure 1."
>
> A1: First, we emphasize that we use the same number (i.e., 100) of training out-of-distribution samples for Figure 1(a)/(b) and 1(c)/(d). As you pointed out, if one increases the number of training out-of-distribution samples for the 50x50 case, Figure 1(b) is expected to be similar to Figure 1(d). In other words, one needs more samples in order to train confidence classifier if samples are generated from the entire space, i.e., 50x50. However, as we mentioned, this might be impossible and not efficient since the number of out-of-distribution training samples might be almost infinite to cover its entire, high-dimensional data space. Therefore, instead, we suggest to sample out-of-distribution close to in-distribution, which could be more effective (given the fixed sampling complexity). The difference in Figure 1 confirms such intuition. We clarified this more in the revision (Section 2.1).
>
> Q2: "Justification of PT."
>
> A2: We agree with you that Zhao et al. did not justify the claim. But, the PT corresponds to the squared cosine similarity of generated samples, and intuitively one can expect the effect of increasing the entropy by minimizing it. In the recent work [1], the authors also used PT to maximize the entropy. However, as we mentioned in A3 for Reviewer_1, after our submission, we actually verified that PT helps, but its gains are relatively marginal in overall. Since PT increases the training complexity, we decided to remove the PT in the revision and have updated all experimental results without using PT. Finally, for interested readers, we also report the effects of PT in the Appendix D. We really appreciate your valuable comments. We updated Section 2.2 and 2.3. Figure 3, 4 and 5. Appendix D. accordingly.
>
> Q3: "About cases where such a strong within-sample assumption may adversely affect performance."
>
> A3: As shown in Table 1 and Table 2 (in Appendix C),  splitting in- and out-of-distributions and optimizing the confidence loss (1) does not adversely affect the classification accuracy due to the high expressive power of deep neural networks in all our experiments. We haven't found a case where our proposed method (based on this assumption) leads to adverse performance. However, more theoretical investigation on whether this assumption guarantees a good performance or whether there is a counterexample would be an interesting future work.
>
> Q4: "How do the modifications affect prediction score calibration?"
>
> A4: Thank you for your great suggestion. After our submission, we actually verified that our method can improve the prediction score calibration. For example, the expected calibration error (ECE) [2] of a classifier trained by our method is lower than that of a classifier trained by the standard cross entropy loss. For interested readers, we reported the corresponding experimental results in the revision (see Appendix C.2).
>
> Q5: "Whether the proposed approach may be more resilient to adversarial attacks."
>
> A5: This is a very interesting question. We believe our method has some potential for being more resilient to adversarial attacks. This is because adversarial examples are special types of out-of-distribution samples. We believe that this should be an interesting future direction to explore.
>
> [1] Shiyu Liang, Yixuan Li, and R Srikant. Principled detection of out-of-distribution examples in neural networks. arXiv preprint arXiv:1706.02690, 2017. (https://arxiv.org/abs/1706.02690)
> [2] Chuan Guo, Geoff Pleiss, Yu Sun, and Kilian Q Weinberger. On calibration of modern neural networks. In ICML, 2017. (https://arxiv.org/abs/1706.04599)
>
> Thanks,
> Authors.

---

### Official Review · AnonReviewer1 · 2017-12-01
**simple, effective method, some discussion/understanding missing**

**Rating:** 6
**Confidence:** 3

**Review:**

This paper proposes a new method of detecting in vs. out of distribution samples. Most existing approaches for this deal with detecting out of distributions at *test time* by augmenting input data and or temperature scaling the softmax and applying a simple classification rule based on the output. This paper proposes a different approach (with could be combined with these methods) based on a new training procedure.

The authors propose to train a generator network in combination with the classifier and an adversarial discriminator. The generator is trained to produce images that (1) fools a standard GAN discriminator and (2) has high entropy (as enforced with the pull-away term from the EBGAN). Classifier is trained to not only maximize classification accuracy on the real training data but also to output a uniform distribution for the generated samples.

The model is evaluated on CIFAR-10 and SVNH, where several out of distribution datasets are used in each case. Performance gains are clear with respect to the baseline methods.

This paper is clearly written, proposes a simple model and seems to outperform current methods. One thing missing is a discussion of how this approach is related to semi-supervised learning approaches using GANS where a generative model produces extra data points for the classifier/discriminator.

 I have some clarifying questions below:
- Figure 4 is unclear: does "Confidence loss with original GAN" refer to the method where the classifier is pretrained and then "Joint confidence loss" is with joint training? What does "Confidence loss (KL on SVHN/CIFAR-10)" refer to?

- Why does the join training improve the ability of the model to generalize to out-of-distribution datasets not seen during training?

- Why is the pull away term necessary and how does the model perform without it? Most GAN models are able to stably train without such explicit terms such as the pull away or batch discrimination. Is the proposed model unstable without the pull-away term?

- How does this compare with a method whereby instead of pushing the fake sample's softmax distribution to be uniform, the model is simply a trained to classify them as an additional "out of distribution" class? This exact approach has been used to do semi supervised learning with GANS [1][2]. More generally, could the authors comment on how this approach is related to these semi-supervised approaches?

- Did you try combining the classifier and discriminator into one model as in [1][2]?

[1] Semi-Supervised Learning with Generative Adversarial Networks (https://arxiv.org/abs/1606.01583)
[2] Good Semi-supervised Learning that Requires a Bad GAN (https://arxiv.org/abs/1705.09783)

---

> ### Author Response · Authors · 2017-12-18
> **Our responses for AnonReviewer1**
>
> We very much appreciate your valuable comments, efforts and times on our paper. We provide our responses for all questions below. Revised parts in the new draft are colored by blue.
>
> Q1: "Figure 4 is unclear."
>
> A1: First, "Confidence loss with original GAN" corresponds to a variant of confidence loss (1) which trains a classifier by optimizing the KL divergence term using samples from a pre-trained original/standard GAN, i.e., GAN generates in-distribution samples. Next, "Joint confidence loss" is the proposed loss (4) optimized by Algorithm 1. Here, we remark that only "Joint confidence loss" optimizes the KL divergence terms using implicit samples from the proposed GAN, i.e., GAN generates "boundary" samples in the low-density area of in-distribution. Finally, "Confidence loss (KL on SVHN/CIFAR-10)" corresponds to the confidence loss (1) using explicit out-of-distribution samples (SVHN or CIFAR-10). For example, "Confidence loss (KL on SVHN)" refers to the method where the KL divergence term in the confidence loss (1) is optimized using SVHN training data. In the revision, we clarified the notations such that the KL divergence term is optimized on samples indicated in the parentheses, i.e., "Confidence loss with original GAN" and "Confidence loss (KL on SVHN/CIFAR-10)" were revised to "Confidence loss (samples from original GAN)" and "Confidence loss (SVHN/CIFAR-10)", respectively. We updated Figure 2 and Figure 4 accordingly.
>
> Q2: "Why does the joint training improve the ability of the model to generalize to out-of-distribution datasets not seen during training?"
>
> A2: It is explained in Section 2.3. In Section 2.1, we suggest to use out-of-distribution samples for training a confident classifier. Conversely, in Section 2.2., we suggest to use a confident classifier for training a GAN generating out-of-distribution samples. Namely, two models can be used for improving each other. Hence, this naturally suggests a joint training scheme in Section 2.3 for confident classifier and the proposed GAN, where both improve as the training proceeds. We emphasize the effect of joint training again in the revision. Please see our revision of Section 2.3 for details.
>
> Q3: "Why is the pull away term necessary and how does the model perform without it?"
>
> A3: We really appreciate your valuable comments.
>
> The pull away term (PT) is not related to "stability." Our intuition was that the entropy of out-of-distribution is expected to be much higher compared to that of in-distribution since the out-of-distribution is typically on a much larger space than the in-distribution. Consequently, we expected that optimizing the PT term is useful for generating better out-of-distribution samples.
> We also note that the PT was recently used [2] for a similar purpose as ours.
>
> However, since we suggest to generate out-of-distribution samples nearby in-distribution (for efficient sampling purpose), its entropy might be not that high and the effect of PT is not clear. After our submission, we actually verified that PT sometimes helps (but not always), and its gains are relatively marginal in overall. Since PT increases the training complexity, we decided to remove the PT in the revision and have updated all experimental results without using PT. Still, for interested readers, we also report the effects of PT in the Appendix D. We updated Section 2.2 and 2.3, Figure 3, 4 and 5, and Appendix D, accordingly.
>
> Q4: "How is this approach related to the semi-supervised approaches in [1][2]? Did you try combining the classifier and discriminator into one model as in [1][2]?"
>
> A4: As briefly mentioned in Section 4, we expect that our proposed GAN might be useful for semi-supervised settings. Also, we actually thought about combining the classifier and discriminator into one model, i.e., adding K+1 class. However, we choose a more "conservative" way to design network architectures so that the original classification performance does not degrade. Extension to semi-supervised learning should be an interesting future direction to explore.
>
> [1] Odena, A. Semi-supervised learning with generative adversarial networks. In NIPS, 2016. (https://arxiv.org/abs/1606.01583)
> [2] Dai, Z., Yang, Z., Yang, F., Cohen, W.W. and Salakhutdinov, R. Good Semi-supervised Learning that Requires a Bad GAN. In NIPS, 2017. (https://arxiv.org/abs/1705.09783)
>
> Thanks,
> Authors

---

### Author Response · Authors · 2017-12-18
**Our common response for all reviewers**

We very much appreciate valuable comments, efforts, and time of the reviewers. We first address a common concern of the reviewers and other issues for each individual one separately.

Q. Comparison between our model and K+1 classes model (within vs. out of distributions model)

A. As all reviewers mentioned, one might want to add the new class for out-of-distribution in the softmax distribution. We actually considered this idea, but did not try since (a) modeling explicitly out-of-distribution can incur overfitting, (b) this prevents us from using the prior inference/detection methods [1,2] (since they do not assume such a new out-of-distribution label) and (c) forcing the softmax distribution be close to uniform is expected to provide additional regularization effects [3]. In other words, we choose a way to design network architectures so that the original classification performance does not hurt and all prior inference algorithms can be still utilized (and further improved under our training method).  As another option, one can try complex density estimators such as PixelCNN [4] to model within vs. out of distributions explicitly, but they are quite difficult to train. Our method is much simpler and easier to use.

We remind that our primary goal is training confident classifiers of standard architectures rather than developing a new architecture for detecting out-of-distribution samples. Please note that this design choice leads to another unexpected advantage: our method can even improve the calibration performance for multi-class classification for in-distribution samples, meaning that a classifier trained by our method can indicate when they are likely to be correct or incorrect for test samples. More specifically, the expected calibration error (ECE) [2] of a classifier trained by our method is lower than that of a classifier trained by the standard cross entropy loss. We also reported the corresponding experimental results in the revision (see Appendix C.2).

For interested readers, we also report experimental results (see Appendix E) to this revision, demonstrating that adding a new class for out-of-distribution is worse than forcing the existing class distribution uniform.

[1] Shiyu Liang, Yixuan Li, and R Srikant. Principled detection of out-of-distribution examples in neural networks. arXiv preprint arXiv:1706.02690, 2017. (https://arxiv.org/abs/1706.02690)
[2] Chuan Guo, Geoff Pleiss, Yu Sun, and Kilian Q Weinberger. On calibration of modern neural networks. In ICML, 2017. (https://arxiv.org/abs/1706.04599)
[3] Pereyra, G., Tucker, G., Chorowski, J., Kaiser, Ł. and Hinton, G. Regularizing neural networks by penalizing confident output distributions. arXiv preprint arXiv:1701.06548, 2017. (https://arxiv.org/abs/1701.06548)
[4] Oord, A.V.D., Kalchbrenner, N., Vinyals, O., Espeholt, L., Graves, A. and Kavukcuoglu, K. Conditional Image Generation with PixelCNN Decoders. In NIPS, 2016. (https://arxiv.org/abs/1606.05328)

Thanks,
Authors

---

### Decision · Program_Chairs · 2018-01-29
**ICLR 2018 Conference Acceptance Decision**

**Decision:**

Accept (Poster)

**Comment:**

Meta score: 6

The paper approaches the problem of identifying out-of-distribution data by modifying the objective function to include a generative term.  Experiments on a number of image datasets.

Pros:
 - clearly expressed idea, well-supported by experimentation
 - good experimental results
 - well-written

Cons:
 - slightly limited novelty
 - could be improved by linking to work on semi-supervised learning approaches using GANs

The authors note that ICLR submission 267 (https://openreview.net/forum?id=H1VGkIxRZ) covers similar ground to theirs.